# Lipocalin-2 Deficiency Reduces Hepatic and Hippocampal Triggering Receptor Expressed on Myeloid Cells-2 Expressions in High-Fat Diet/Streptozotocin-Induced Diabetic Mice

**DOI:** 10.3390/brainsci12070878

**Published:** 2022-07-02

**Authors:** Hyun Joo Shin, Zhen Jin, Hyeong Seok An, Gyeongah Park, Jong Youl Lee, So Jeong Lee, Hye Min Jang, Eun Ae Jeong, Kyung Eun Kim, Jaewoong Lee, Dae Young Yoo, Gu Seob Roh

**Affiliations:** 1Department of Anatomy and Convergence Medical Science, Institute of Health Science, College of Medicine, Gyeongsang National University, Jinju 52727, Korea; k4900@hanmail.net (H.J.S.); gudtjr5287@hanmail.net (H.S.A.); jyv7874v@naver.com (J.Y.L.); thwjd5411@naver.com (S.J.L.); gpals759@naver.com (H.M.J.); jeasky44@naver.com (E.A.J.); kke-jws@hanmail.net (K.E.K.); woongs1111@gmail.com (J.L.); dyyoo@gnu.ac.kr (D.Y.Y.); 2Department of Anatomy and Neurobiology, College of Medicine, University of Tennessee Health Science Center, Memphis, TN 38163, USA; zkim777@gmail.com (Z.J.); imkapark@gmail.com (G.P.)

**Keywords:** lipocalin-2, TREM2, inflammation, diabetic mouse

## Abstract

Background: Lipocalin-2 (LCN2) is an acute-phase protein that has been linked to insulin resistance, diabetes, and neuroinflammatory diseases. Triggering receptor expressed on myeloid cells-2 (TREM2) has been also implicated in microglia-mediated neuroinflammation. However, the potential role of LCN2 on TREM2 in diabetic mouse models is not fully understood. Methods: We investigated hepatic and hippocampal TREM2 expressions in high-fat diet (HFD) and streptozotocin (STZ)-induced diabetic LCN2 knockout (KO) mice. Results: In addition to increased serum LCN2 level, diabetic wild-type (WT) mice had insulin resistance and hepatic steatosis. However, LCN2 deletion attenuated these metabolic parameters in diabetic mice. We also found that LCN2 deletion reduced hepatic inflammation and microglial activation in diabetic mice. In particular, diabetic LCN2 KO mice had a reduction in hepatic and hippocampal TREM2 expressions compared with diabetic WT mice. Furthermore, we found that many TREM2-positive Kupffer cells and microglia in diabetic WT mice were reduced through LCN2 deletion. Conclusions: These findings indicate that LCN2 may promote hepatic inflammation and microglial activation via upregulation of TREM2 in diabetic mice.

## 1. Introduction

Lipocalin-2 (LCN2), which is secreted by various cell types, is associated with several metabolic diseases and brain injuries such as ischemic brain disease, diabetic encephalopathy, and Alzheimer’s disease (AD) [1,2,3,4]. Despite the fact that LCN2 was initially isolated from neutrophil granules, it is functionally linked to inflammation due to a nuclear factor-κB (NF-κB) binding site in the promoter region of the LCN2 gene [5,6]. Many lines of evidence reported that LCN2 is primarily expressed in astrocytes in neuroinflammatory diseases [4,7,8]. In addition, because the LCN2 receptor is expressed on microglia, its activation induces microglial polarization toward the M1 pro-inflammatory state, mediated via NF-κB activation [4,9]. Our recent study showed that LCN2 promoted neuroinflammation and memory impairments in a high-fat diet (HFD)/streptozotocin (STZ)-induced diabetic mouse model [10]. It has been reported that LCN2 is associated with microglial activation and neuroinflammation in a mouse model of amyloid-beta oligomer-induced AD [11]. However, further role of LCN2 in microglial activation in diabetic mice is necessary.

The triggering receptor expressed on myeloid cells-2 (TREM2) is a cell surface receptor primarily expressed in macrophage, osteoclast, and microglia that belongs to the immunoglobulin superfamily [12,13,14]. In particular, microglia are brain-resident macrophages that serve as the brain’s first and primary immune defense mechanism. Microglia play a critical role in neuroinflammation in response to brain damage and typically undergo rapid morphological and functional changes against injuries [15]. Moreover, a recent study has reported that TREM2 regulates high-glucose-induced inflammation in BV2 microglial cells [16]. In a transgenic mouse model of AD, TREM2 deletion exacerbated neuroinflammation via Toll-like receptor 4 (TLR4)-mediated mitogen-activated protein kinase (MAPK) signaling pathway [17]. On the other hand, TREM2 deficiency accelerates the progression of nonalcoholic fatty liver disease (NAFLD), whereas TREM2 overexpression has a protective effect on NAFLD [18]. Based on these bodies of evidence, we suggest that TREM2 appears to play paradoxical roles in response to various stimuli. However, the role of LCN2 deletion in hepatic and hippocampal TREM2 expression in HFD-/STZ-induced diabetic mice is still unclear.

Here, we hypothesize that macrophage-derived Kupffer cells and microglia express TREM2 in the liver and hippocampus of HFD/STZ-induced diabetic mice. The current study supports the notion that LCN2 may promote hepatic inflammation and microglial activation via upregulation of TREM2 expression in diabetic mice.

## 2. Materials and Methods

### 2.1. Animals

Male C57BL/6J and LCN2 knockout (KO) mice were purchased from Central Laboratory Animal, Inc. (Seoul, South Korea) and The Jackson Laboratory (Bar Harbor, ME, USA), respectively. Wild-type (WT) and LCN2 KO mice were back-crossed for 8–10 generations on the C57BL/6J background to reduce background phenotypic effects. Polymerase chain reaction analysis of genomic DNA verified the lack of LCN2. All animal experiments at Gyeongsang National University were conducted in accordance with approved animal protocols and the guidelines established by the Animal Care Committee (No. GNU-190701-M0033). A diabetic mouse model was generated by feeding with HFD (60 kcal% fat; Research Diets, New Brunswick, NJ, USA) and STZ (Sigma-Aldrich, St Louis, MO, USA) treatment, as previously described [19]. Briefly, 5-week-old male WT or LCN2 KO mice (*n*  =  8 mice per group) were fed an HFD for 20 weeks. After 16 weeks on HFD, mice were injected with a single intraperitoneal STZ dose (100 mg/kg). As control (CTL) mice, WT or LCN2 KO mice (*n*  =  7 mice per group) were fed a standard chow diet and injected with the sodium citrate buffer vehicle. All mice were sacrificed at 25 weeks.

### 2.2. Metabolic Parameters

In order to collect blood samples from the left ventricle of the heart, mice were given the anesthetic Zoletil (5 mg/kg, Virbac Laboratories, Carros, France). Serum glucose, aspartate aminotransferase (AST), and alanine aminotransferase (ALT) were measured in the obtained supernatants by the Green Cross Reference Laboratory (Youngin-si, Korea).

### 2.3. Enzyme-Linked Immunosorbent Assay (ELISA)

Serum insulin or LCN2 (*n* = 7–8 mice per group) were measured using a mouse insulin ELISA kit (Shibayagi Co., Gumma, Japan) or mouse LCN2 ELISA kit (R&D Systems, Minneapolis, MN, USA) according to the manufacturer’s instructions, respectively.

### 2.4. Hematoxylin and Eosin (H&E) and Nile Red Staining

To measure the pancreatic islet area, paraffin-embedded pancreas total sections (5 µm) were stained with H&E, and percentages were obtained using i-Solution (IMT i-Solution Inc., Vancouver, BC, Canada) in three fields (150 × 150 µm^2^) randomly selected from two continuous sections (*n* = 4 per group). To determine the NAFLD activity score, paraffin-embedded liver sections (5 µm) were deparaffinized, stained with H&E, and subjected to histopathological examination using a BX51 light microscope (Olympus, Tokyo, Japan). The NAFLD activity score was quantified as the sum of the scores for steatosis (0–3), lobular inflammation (0–2), and hepatocellular ballooning (0–2) [20]. Nile red (Sigma-Aldrich), a fluorescent lipid droplet dye, was applied to assess hepatic lipid accumulation in HFD-/STZ-induced diabetic model mice. Nile red (Sigma-Aldrich) was used to stain frozen liver sections (5 µm) for 10 min. After washing, the sections were counterstained with Mayer’s hematoxylin (Sigma-Aldrich) for 45 s. The fluorescence intensity of each section was measured at 594 nm using a BX51-DSU microscope (Olympus).

### 2.5. Western Blotting

The liver and hippocampi (*n*  =  3–4 mice per group) were quickly extracted and frozen. As previously described, total lysates and nuclear fractions were prepared as previously described [21]. Briefly, the liver and hippocampi were homogenized in a T-PER Tissue Protein Extraction Reagent (Thermo Fisher Scientific, Waltham, MA, USA) for protein extraction and in a NE-PER Nuclear and Cytoplasmic Extraction Reagent (Pierce, Rockford, IL, USA) to obtain nuclear fractions. Standard methods were used to conduct the Western blot analyses. Proteins were immunoblotted with LCN2 (AF1857, R&D Systems), tumor necrosis factor-α (TNF-α; ab9739, Abcam, Cambridge, MA, USA), NF-κB p65 (#6956, Cell Signaling, Danvers, MA, USA), and TREM2 (sc-373828, Santa Cruz Biotechnology, Dallas, TX, USA) antibodies. To normalize total and nuclear fraction protein levels, loading controls (β-actin (A5441, Sigma-Aldrich) and p84 (ab487, Abcam), respectively) were used. Each target protein and internal control were probed and visualized via enhanced chemiluminescence (Pierce), and band densitometry was performed using the Multi-Gauge V 3.0 image analysis program (Fujifilm, Tokyo, Japan).

### 2.6. Immunofluorescence

Deparaffinized liver sections or frozen brain sections were incubated overnight with antibodies against F4/80 (sc-71085, Santa Cruz), Ly6G (ab25377, Abcam), and TREM2 (sc-373828, Santa Cruz) at 4 °C. After washing with 0.1 M PBS, sections were incubated with Alexa Fluor 488- and 594-conjugated donkey secondary antibodies (Invitrogen Life Technologies, Carlsbad, CA, USA). Nuclei were stained with 4’, 6-diamidino-2-phenylindole (DAPI; 1:20,000, Invitrogen). The fluorescence intensity of the solution was measured using a BX51-DSU microscope (Olympus). For the measurement of the intensity of immunostained protein from liver or brain sections, 12–16 fields (120 × 120 µm^2^ or 20 × 20 µm^2^) were randomly selected from each section (*n* = 4 per group) and measured with i-Solution (IMT i-Solution Inc.).

### 2.7. Statistical Analysis

PRISM 7.0 (GraphPad Software Inc., San Diego, CA, USA) was used to conduct statistical analyses. Two-way analysis of variance (ANOVA), followed by Tukey’s post hoc test, was used to evaluate group differences. All values are expressed as the mean ± the standard error of the mean (SEM). Significance was defined as a *p*-value less than 0.05.

## 3. Results

### 3.1. Effects of LCN2 Deletion on Insulin Resistance in HFD/STZ-Induced Diabetic Mice

To investigate whether LCN2 deletion inhibits insulin resistance, we used HFD/STZ-induced diabetic WT and LCN2 KO mice. Diabetic WT and LCN2 KO mice had higher body weights than their non-diabetic counterparts, and both body weight and fasting glucose levels in diabetic LCN2 KO mice were significantly lower than those in diabetic WT mice (Figure 1A,B). As STZ induces diabetes through the selective destruction of pancreatic islet cells [22], no significant differences in serum insulin levels were found between groups (Figure 1C). Diabetic WT mice had significantly smaller pancreatic islet areas than WT CTL mice, but this effect was reversed in diabetic LCN2 KO mice (Figure 1D,E). Furthermore, we found that circulating LCN2 levels were increased in diabetic WT mice, compared with WT CTL mice (Figure 1F). These results indicate that LCN2 deletion improves hyperglycemia in diabetic mice.

### 3.2. Effects of LCN2 Deletion on Hepatic Steatosis in HFD/STZ-Induced Diabetic Mice

Next, we investigated whether LCN2 deletion attenuated hepatic steatosis in diabetic mice. Diabetic LCN2 KO mice had significantly lower liver weights than diabetic WT mice, corresponding to differences in body weight (Figure 2A). We found that increased circulating AST and ALT levels in diabetic WT mice were reduced by LCN2 deletion (Figure 2B). H&E-stained histological analysis revealed that diabetic LCN2 KO mice exhibited lower NAFLD activity scores, a surrogate for liver damage, than diabetic WT mice (Figure 2C,D). Nile red staining showed fewer areas containing lipid droplets in diabetic LCN2 KO mice than in diabetic WT mice (Figure 2E).

### 3.3. Effects of LCN2 Deletion on Hepatic TREM2 Expression in HFD/STZ-Induced Diabetic Mice

Given that LCN2 KO mice had hepatic inflammation in diabetic mice, we determined hepatic TNF-α expression and Kupffer cell infiltration (Figure 3). In line with circulating LCN2 levels, Western blot analysis showed that HFD/STZ treatment increased hepatic LCN2 expression (Figure 3A). Increased hepatic TNF-α and TREM2 expressions in diabetic mice were significantly inhibited by LCN2 deletion (Figure 3A,B). Furthermore, double immunofluorescent staining revealed that many TREM2-positive cells were co-localized with F4/80-positive Kupffer cells or Ly6G-positive neutrophils in diabetic WT mice, but these positive cells were reduced by LCN2 deletion (Figure 3C–E). Taken together, these findings indicate that LCN2 deletion could attenuate hepatic steatosis and inflammation in diabetic mice.

### 3.4. Effects of LCN2 Deletion on Hippocampal TREM2 Expression in HFD-/STZ-Induced Diabetic Mice

Microglia are a specialized population of macrophage-like cells capable of triggering a robust inflammatory response and serve as immune sentinels in the brain [23]. To investigate whether LCN2 deletion inhibited diabetic neuroinflammation, we determined microglial activation in the hippocampus of diabetic mice. In addition to increased hippocampal LCN2 protein, TREM2 and nuclear NF-kBp65 protein levels were significantly increased in the hippocampus of diabetic WT mice, whereas these protein expressions were completely reversed by LCN2 deletion (Figure 4A,B). For measuring microglial activation by assessing microglia-positive Iba-1 and TREM2 co-localization, we performed double immunofluorescence (Figure 4C). We found that many TREM2-positive cells were co-localized with Iba-1-positive microglia in diabetic WT mice, but these cells were reduced by LCN2 deletion (Figure 4D). Thus, these results indicate that LCN2 deletion could reduce microglial activation in the diabetic hippocampus via inhibiting TREM2.

## 4. Discussion

Many studies have reported that LCN2 functions as proinflammatory adipocytokine involved in obesity-related metabolic complications and inflammatory diseases. In particular, LCN2 is expressed in various cells including adipocytes, hepatocytes, Kupffer cells, macrophages, neutrophils, and astrocytes. LCN2 appears to play paradoxical roles in NAFLD and sepsis [24,25]. In diabetic encephalopathy, elevated LCN2 promotes neuroinflammation and oxidative stress [11]. However, in response to different stimuli and exposed time, specific LCN2-derived cells affect neuronal damage or activated astrocyte, and microglial activation. On the other hand, as with the dual role of LCN2, TREM2 also has paradoxical functions in some inflammatory diseases. Therefore, in the present study, we attempted to evaluate the potential role of TREM2 in diabetic LCN2 KO mice. As expected, we found that LCN2 deletion reduced insulin resistance and hepatic steatosis. Notably, our findings showed that myeloid-derived Kupffer cells and microglia-positive cells were co-localized with TREM2-stained cells in the liver and hippocampus of diabetic mice, and these increased TREM2 expressions were reduced by LCN2 deletion. Taken together, these results suggest that LCN2 may contribute to the activation of macrophages and promote the progression of NAFLD and neuroinflammation in diabetic mice.

In accordance with the fact that circulating LCN2 level is increased in insulin resistance and type 2 diabetes [26], its serum level was also elevated in HFD/STZ-induced diabetic mice. LCN2-deficient mice revealed lower body weight and serum glucose than diabetic WT mice. Although there was no significant change in serum insulin levels, histological analysis showed a small area of a pancreatic islet in HFD/STZ-treated mice compared with control mice. However, this small area of the pancreatic islet was inhibited by LCN2 deletion. Thus, these findings indicate that elevated LCN2 may be associated with insulin resistance.

Our previous studies have revealed that caloric restriction reduced increased LCN2 levels in genetic ob/ob and db/db mice with hepatic steatosis [27,28]. In the present study, diabetic LCN2 KO mice also revealed lower body weight and NAFLD score activity, compared with diabetic WT mice. Consistent with our result, LCN2 mediates non-alcoholic steatohepatitis (NASH) by promoting neutrophil–macrophage crosstalk via the induction of chemokine (C-X-C motif) receptor 2 [29]. As shown in Figure 2D, HFD/STZ-induced hepatic inflammation was significantly reduced by LCN2 deletion. It supports the finding that LCN2 deletion reduced the increase in hepatic TNF-α expression and F4/80-positive Kupffer cells in diabetic mice. TREM2 is gradually increased in the adipose tissue of HFDfed dogs and diabetic db/db mice [30,31]. Although TREM2 has predominantly been known as a negative regulator of the inflammatory response in macrophages [32], another study has reported that TREM2 amplifies the inflammatory response induced by the TLR4 pathway [33]. Park et al. demonstrated that HFD-fed TREM2 transgenic (TG) mice manifested adipocyte hypertrophy, insulin resistance, and hepatic steatosis [34]. They suggested that HFD-fed TG mice induce the production of cytokines, leading to the recruitment of macrophages into adipose tissue. In the present study, we also showed that, in addition to increased hepatic TREM2 protein, many TREM2-positive cells were observed in F4/80-positive Kupffer cells and neutrophils in diabetic WT mice compared with control mice. These data indicate that, in elevated LCN2 levels, TREM2 overexpression may function to regulate immune response against hepatic steatosis and inflammation. We hypothesize that inhibiting immune response by LCN2 deletion may cause the inhibition of macrophage infiltration and that TREM2 may have a role in the inflammatory response in NAFLD of diabetic mice.

In our previous studies, hippocampal LCN2 protein increased in several obese and diabetic mice as well as serum LCN2 levels [1,27]. Furthermore, in line with the lines of evidence revealing that elevated LCN2 levels are closely linked to memory deficits in diabetic mice or patients [10], increased hippocampal LCN2 expression is correlated with neuroinflammation and cognitive impairment. It supports the result that caloric-restriction-treated ob/ob mice revealed enhanced memory deficits with reversed serum and hippocampal LCN2 proteins [27]. In our recent study, we demonstrated that secreted LCN2 increased inflammatory cytokines and NF-κBp65 high-glucose-treated mouse hippocampal HT22 cells [10]. As downregulation of nuclear NF-κBp65 in the hippocampus in diabetic LCN2 KO mice, increased TREM2 levels were also inhibited by LCN2 deletion. Our finding that microglial activation was increased in diabetic WT mice is consistent with the finding that many Iba-1-positive microglia are observed in the diabetic hippocampus by upregulating TREM2 [35]. However, TREM2 signaling appears to have both pro- and anti-inflammatory effects in AD [36]. Increased TREM2 expression may prevent AD progression during early stages, but several TREM2 variants have been identified during later stages of AD, resulting in brain damage [13]. Conversely, the hippocampal overexpression of TREM2 attenuated microglial activation and cognitive impairment in mice fed with HFD for 50 weeks [37]. In contrast to the microglial effect, decreased astrocytic TREM2 levels have demonstrated beneficial effects on learning and memory in aged mice [38]. The overexpression of TREM2 using an adeno-associated viral vector markedly reduced the activation of the NF-κB pathway and the associated neuroinflammatory response, affecting the levels of interleukin-1β, TNF-α, TLR4, and inducible nitric oxide synthase in the hippocampus of HFD-fed mice [38]. Thus, our study elucidated the idea that macrophage-derived TREM2 may be involved in the inflammatory response in the diabetic brain. Furthermore, we suggest that the potential molecular mechanisms underlying the anti- and pro-inflammatory effects of TREM2 in diabetes may require further study.

## 5. Conclusions

These findings show that LCN2 deletion reduces HFD/STZ-treated insulin resistance, hepatic steatosis, and microglial activation. LCN2 regulates hepatic and hippocampal inflammation via macrophage-derived TREM2 expression. This leads to the hypothesis that TREM2 may be a potential target for the treatment of diabetes-associated inflammation by establishing a connection between LCN2-mediated neuroinflammation and microglial activation. Further research remains necessary to elucidate the biological relevance of LCN2-linked TREM2 in diabetes.

## Figures and Tables

**Figure 1 brainsci-12-00878-f001:**
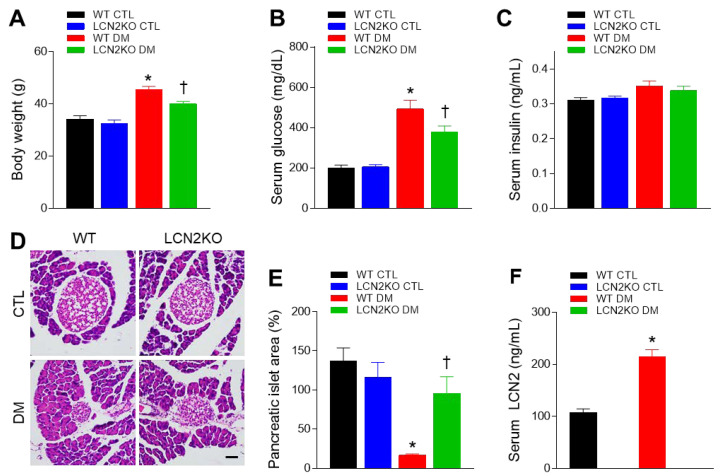
Effects of LCN2 deletion on insulin resistance in HFD/STZ-induced diabetic mice: (**A**) body weights, (**B**) fasting serum glucose levels, and (**C**) serum insulin levels for each group; (**D**) representative images showing H&E-stained pancreatic sections; scale bar = 50 µm; (**E**) pancreatic islet areas measured for each group; (**F**) serum LCN2 levels measured for each group. CTL: control; DM: diabetic mice. Data are presented as the mean ± SEM. The indicated *p*-values represent a two-way ANOVA, followed by Tukey’s post hoc test. * *p* < 0.05 vs. WT CTL. ^†^ *p* < 0.05 vs. WT DM.

**Figure 2 brainsci-12-00878-f002:**
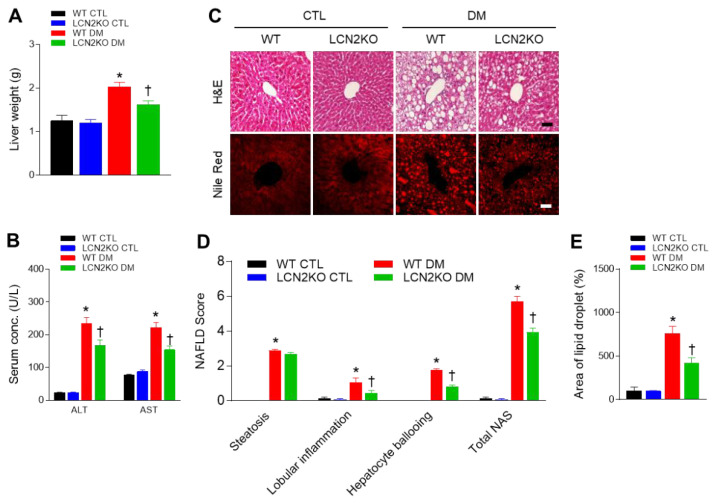
Effects of LCN2 deletion on hepatic steatosis in HFD/STZ-induced diabetic mice: (**A**) liver weight and (**B**) serum AST and ALT activity levels for each group; (**C**) representative images showing H&E and Nile red staining of liver sections; scale bar = 50 µm; (**D**) histological NAFLD scores for each group; (**E**) percentages of Nile red-positive areas for each group; scale bar = 10 µm. CTL: control; DM: diabetic mice. Data are presented as the mean ± SEM. The indicated *p*-values represent a two-way ANOVA, followed by Tukey’s post hoc test. * *p* < 0.05 vs. WT CTL. ^†^ *p* < 0.05 vs. WT DM.

**Figure 3 brainsci-12-00878-f003:**
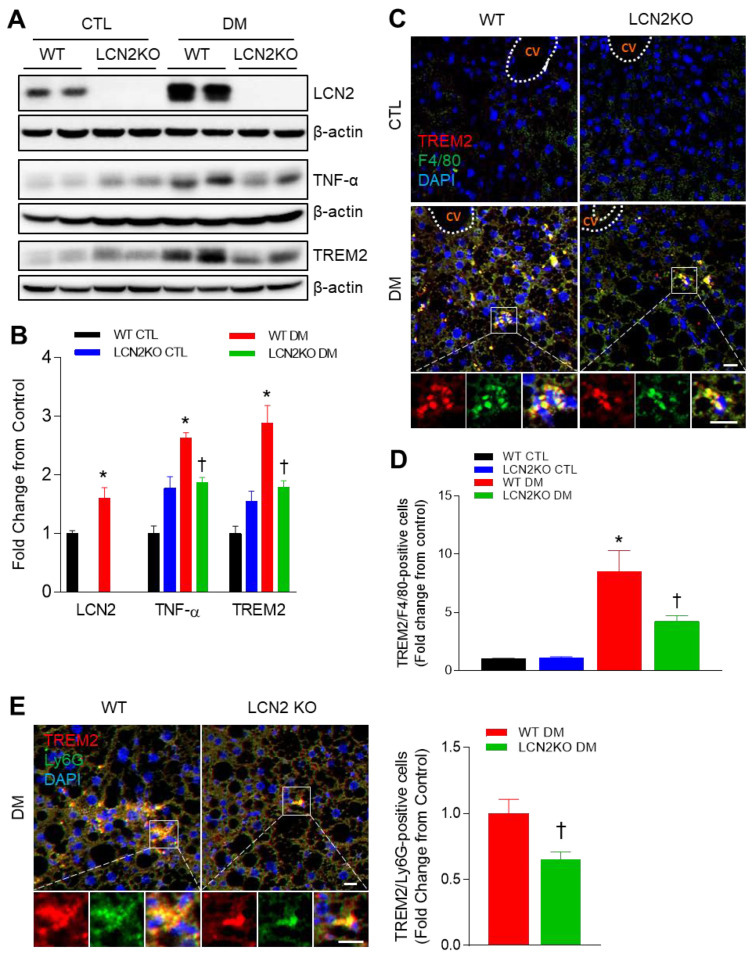
Effects of LCN2 deletion on hepatic TREM2 expression in HFD/STZ-induced diabetic mice: (**A**,**B**) Western blot analysis and quantitation of LCN2, TNF- α, and TREM2 proteins in liver lysates. β-actin was used as loading control; (**C**) representative immunofluorescence images of TREM2 (red) and F4/80 (green) in liver sections; CV: central vein; (**D**) quantification of co-localized TREM2 and F4/80-immunostained cells in the images; (**E**) representative immunofluorescence images of TREM2 (red) and Ly6G (green) in liver section; quantification of co-localized TREM2 and Ly6G-immunostained cells in the images; scale bar = 10 µm. DAPI (blue) was used to stain nuclei. CTL: control; DM: diabetic mice. Data are presented as the mean ± SEM. The indicated *p*-values represent a two-way ANOVA, followed by Tukey’s post hoc test. * *p*  <  0.05 vs. WT CTL. ^†^ *p* < 0.05 vs. WT DM.

**Figure 4 brainsci-12-00878-f004:**
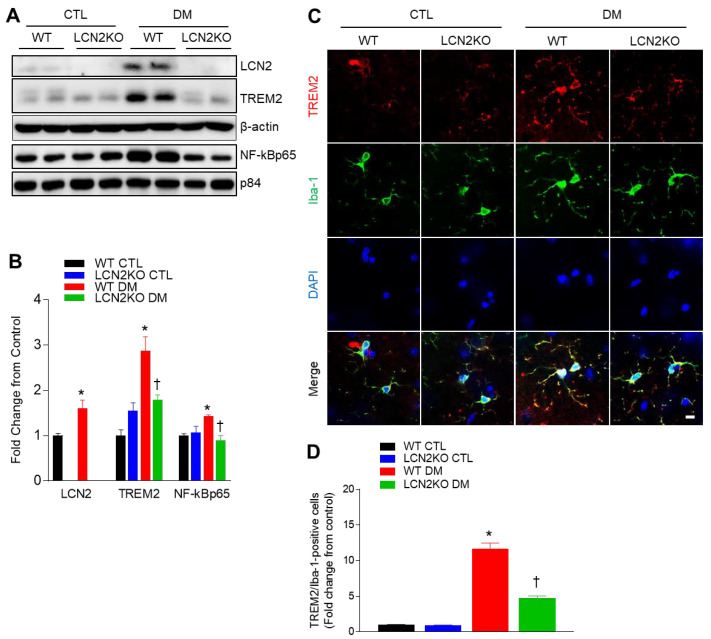
Effects of LCN2 deletion on microglial activation and hippocampal TREM2 expression in HFD/STZ-induced diabetic mice: (**A**,**B**) Western blot analysis and quantification of LCN2, TREM2, and NF-κBp65 in the hippocampus; β-actin and p84 were used as loading controls for total protein and nuclear protein, respectively; (**C**) representative immunofluorescence images of TREM2 (red) and Iba-1 (green) in hippocampal sections; DAPI (blue) was used to stain nuclei; scale bar = 10 µm; (**D**) quantification of co-localized TREM2 and Iba-1-immunostained cells in the images. CTL: control; DM: diabetic mice. Data are presented as the mean ± SEM. The indicated *p*-values represent a two-way ANOVA, followed by Tukey’s post hoc test. * *p*  <  0.05 vs. WT CTL. ^†^ *p* < 0.05 vs. WT DM.

## Data Availability

Data are contained within the article.

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
