# Peer review of "Lipocalin-2 Deficiency Reduces Hepatic and Hippocampal Triggering Receptor Expressed on Myeloid Cells-2 Expressions in High-Fat Diet/Streptozotocin-Induced Diabetic Mice"

_brainsci, 2022, doi:10.3390/brainsci12070878_

Round 1
Reviewer 1 Report
In the study entitled « Lipocalin-2 deficiency reduces hepatic and hippocampal triggering receptor expressed on myeloid cells-2 expressions in high-fat diet/streptozotocin-induced diabetic mice” Hyun Joo Shin et al induced diabetes using combined treatment by High fat Diet (HFD) and streptozotocin (STZ) in WT or Lcn2 KO animals. They investigated the impact of Lcn2 expression on metabolic markers of diabetes as well as liver and hippocampal inflammation. They observe that Lcn2 invalidation protects from HFD/STZ deleterious effect in the pancreas as well as in the liver. They further report that Lcn2 invalidation decreases TREM positive kupffer cells and microglia in the liver and hippocampus respectively.
While the Experiment design and the results are clear, I have comments:
1-in Figure 1 it is rather surprising that 9 weeks after STZ injection mice in WT group have such a high level of serum insulin (Figure 1C) while the pancreatic islet area measurement is showing an almost complete disappearance of beta cells mass.
2- Considering the recent reports about the role of Lcn2 in feeding behavior, it seems mandatory in an HFD model to report the impact of Lcn2 KO on the food intake in order to discriminate a direct effect of Lcn2 KO on the metabolic/physiopathologic processes rather than a decrease in the HFD efficacy.
3- The specificity of the effect of Lcn2 on TREM2 positive population needs to be better established. The authors should measure the impact of Lcn2 invalidation on the overall population of macroglia/kupffer cells in order to demonstrate if TREM2 expression is induced by Lcn2 in resident macrophage or if it’s an overall increase of the population.
4-The fact that the authors described a multi-organ phenomenon suggest that the effect of Lcn2 depends on its secretion. Nevertheless, it needs confirmation using recombinant Lcn2 in HFD/TZ KO Lcn2 mice. Indeed, Lcn2 can be expressed in macrophage and may be involved directly in the TREM2 positive cells for TREM2 induction.
5-The authors limited their Lcn2 dependent TREM2 induced expression study to the kupffer cells and the microglia. It will be of interest to investigate if serum Lcn2 induced by HFD/STZ can remotely induce TREM2 expression in other organ such as intestine (dendritic cells expressing TREM2) to test whether is it a pan organ effect or if there is a specificity.
6- In the end of the introduction, the author state: “The current study supports the notion that LCN2 may promote hepatic inflammation and microglial activation via downregulation of TREM2 expression in diabetic mice.” I guess that they meant upregulation.
Minor comments:
Authors should check for typos in the manuscript.
Reviewer 2 Report
-The authors present an analysis of the effects and interactions of LCN2 and TREM2 on diabetic and liver markers in a diabetic mouse model. The study is well-performed and the write up is comprehensive and nicely done. A few minor points to consider.
-This is a nice introduction. Succinct, to the point. One of the better introductions I have seen for a while.
-were the ELISAs performed in replicates?
-in discussion, the paradoxical roles of LCN2 - Could this be due to tissue-specific roles/effects?
-given your previous work with LCN2 (diet), this is an interesting extension. Could future work involve investigating the response of LCN2 after anti-diabetic medications? Lifestyle (exercise)?
-Is there potentially an "LCN inducer" that could be used in future studies to probe the mechanisms you are investigating?
Round 2
Reviewer 1 Report
The authors replies to my comments fully and added interesting new data. I have no further concerns.